# The Amazon Forest Preservation by Clarifying Property Rights and Potential Conflicts: How Experiments Using Fit-for-Purpose Can Help

**Bastiaan Reydon** [1,*,†], **Mathilde Molendijk** [1], **Nicolas Porras** [1,‡] **and Gabriel Siqueira** [2]

1   Kadaster International Cadastre, Land Registry and Mapping Agency (Kadaster), P.O. Box 9046, 7300 GH Apeldoorn, The Netherlands; Mathilde.molendijk@kadaster.nl (M.M.); nicolas.porras@landinpeace.com (N.P.)
2   Institute of Economics, State University of Campinas, Campinas 13083-857, Brazil; G210058@dac.unicamp.br
*   Correspondence: bastiaan.reydon@kadaster.nl; Tel.: +31-088-1832200
†   This author is in Brazil.
‡   This author is in Colombia.

**Abstract:** The burning and the deforestation of the Brazilian Amazon forest, which has been recently highlighted by the international press and occurs mostly on public or undesignated land, calls for an in-depth examination. This has traditionally been the main way to grab land, speculate, and simultaneously prove ownership by its occupation. The absence of mapping, registration, and an effective regulation of land property in Brazil, particularly in the Amazon, plays an important role in its deforestation. Recent estimations, besides others, show that the amount of land in this condition is around 200 million ha, near enough $\frac{1}{4}$ of the national surface. This article, besides examining the Brazilian deforestation characteristics, provides evidence that clear landholders' rights diminishes deforestation, and that proposals based on concrete cases of participatory clarification of land rights in forest regions using fit for purpose (FfP) methodology promote forest preservation. The article finishes with an example of a land rights clarifying case from small, medium, large, and traditional population landholders. The case is important to illustrate that it is possible to clarify land rights in a FfP way and how that increases the security of landholders, diminishing the pressure on the land and thus reducing the potential deforestation.

**Keywords:** Amazon; deforestation; Fit-For-Purpose land administration; participatory mapping

## 1. Introduction

The burning and deforestation of the Brazilian Amazon forest, which has recently been highlighted by the international press, play an important role in the global climate equilibrium and on global greenhouse gas (GHG) emissions, an important aspect of the UN's Sustainable Development Goals (SDG). With Bolsonaro's administration, there was an evident escalation of deforestation in Brazil, which stimulated the discussion around it[1]. Along with the recent dismantling of environmental policies, the government proposed important changes in the legislation regarding land regularization that could increase the possibility to grab land, and thereby also incentivize deforestation (see Kluck (2020) [2] for details). Not only because of the parliament and pressure from social movements, but it was possible to avoid further damage[2], given a coordinated effort that also raised awareness regarding the legal undefinition of land as an important driver of the current deforestation. Due to this, many seek concrete solutions for the regularization of these

---

1   The article in Science by Escobar (2020) [1] shows that since the beginning of Bolsonaro's government, forest protection policies have diminished and deforestation has increased.
2   There were many public discussions, technical publications, and political motions in Brazil regarding the Provisional Measure (*Medida Provisória*) no. 910 and later Law Proposal no. 2633 during the year 2020.

troubling issues in the Amazon region. This article aims to provide a solution for these as it proposes the use of a participatory fit-for-purpose (FfP) approach to clarify land rights in the Amazon region.

In the literature, there is a perception that deforestation usually occurs when land is grabbed or bought to be used immediately or in the long run, as evidenced by Reydon (2011) [3]. Deforestation occurs as it creates revenues from logging, crops, cattle ranching, land appreciation, and ultimately, deforestation is necessary to prove or assure ownership. The absence of a cadaster, as efficient registration, and an effective regulation of land property in Brazil, especially in the Amazon region, contributes to deforestation as an attractive venture, as shown by Reydon et al. (2019) [4] and others.

Since the beginning of the 2000s, mainly based on the Constitution of 1988 and other specific policies, Brazil has created numerous protected areas for its indigenous people and for environmental purposes, summing about 205.8 million hectares. The indigenous reservations and the protected areas represent 24.2 % of the Brazilian surface[3] and are the ones that mostly protect the forests, and after the New Forest Code of 2012 (*Código Florestal*), there were expectations for the diminishing of deforestation.

With the Forest Code was created the CAR (Cadastro Ambiental Rural), a land use georeferenced mapping system, to monitor the forested areas in private properties. It is an opensource dataset that made possible many studies on the deforestation and patterns of forests maintenance on private properties. Two important examples are Alix-Garcia et al. (2017) [6] and L'Roe, J et al. (2016) [7], who, in different ways, showed that this cadastral system plays an important role in monitoring the deforestation on private properties.

On the other hand, Moutinho et al. (2016) [8] showed that much of the deforestation occurs on public land or undesignated land[4], but could not be evidenced only by the CAR data, as its dataset is focused on presumed ownership of georeferenced areas. Based not only on evidence from Brazil, but Robinson et al. (2014) [9] also showed, based on international literature, that clear land rights have a decisive role in preserving forests, especially in Latin America.

As most studies on deforestation have dealt with private properties, the main aim of this article is to emphasize that, besides the efforts of reducing deforestation on private properties, there is an urgent need to clarify land rights in general, but more intensively on public and undesignated land. This article will not only present the characteristics of deforestation and the consequences of the lack of land administration in Brazil, but will also show a concrete example of actions that clarified land rights and avoided conflict around land in forested areas in the Amazon region.

Therefore, this article will be divided into four items:

(a)   Deforestation in the Brazilian Amazon: quantification, importance, and characteristics;
(b)   Evidence of the relation between deforestation and lack of clear property rights in Brazil and in the Amazon region;
(c)   Why good land administration reduces deforestation; and
(d)   A case using a participatory fit-for-purpose approach to help clarify land rights in a forested region.

By presenting a case study, it is expected to highlight methodologies that can clarify landholder ownership and other traditional population landholder rights, but also contribute to diminish potential conflicts over undesignated public land. As a result, the experiences were conducted to find ways to improve the legislation and the institutional settings, so that land rights can be clarified in an easier/affordable way and can help maintain the Amazon rainforest.

---

[3]   For details, see the amount of land for each category in Sparovek et al. (2019) [5].
[4]   Sparovek et al. (2019) [5] estimates, based on all Brazilian available spatial mapping efforts, that the country has 196,056 million ha with no clear destination as they are undesignated or unregistered land that are the most vulnerable for grabbing and deforestation.

## 2. Amazon Rainforest Deforestation

Figure 1 demonstrates that the deforestation in the Amazon region in recent years is of 69 to 11.1 thousand km² a year[5], based on satellite images. This is a lot less than previous decades, but is still a high level of deforestation for a biome like the Amazon, considering its biodiversity and its role in regulating the world's climate and especially rainwater, which is extremely important for good agriculture productivity without irrigation in the Central and Southern regions of Brazil[6].

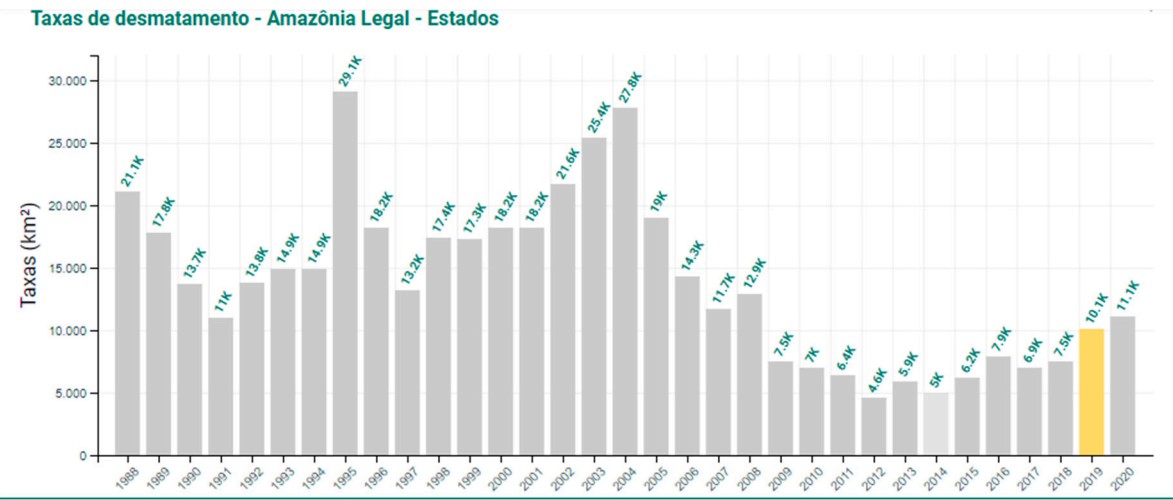

**Figure 1.** Annual deforestation in the Legal Amazon (Km² a year) [11].

Numerous studies[7] have evaluated the causes of deforestation in the Brazilian Amazon. One of them, by Moutinho et al. (2016) [8], lists the six main factors as follows: (a) The growth acceleration plan (PAC) and infrastructure constructions; (b) growth in the demand for commodities (meat and grain); (c) unsustainable policy on rural settlements (Agrarian Reform); (d) inadequate application of the Forest Code; (e) lobbying by agribusiness in the National Congress; and (f) land ownership ambiguities and the existence of undesignated public forests. In more general terms, Margulis (2003) [12] states that the main drivers of deforestation are:

a.  Increase in profits linked to the use of land in the Amazon;
b.  Accessibility of public policies and loans for the region;
c.  Installation of infrastructure for access to frontier areas; and
d.  Phases of GDP growth.

While not disagreeing with the aforementioned conclusions, Reydon et al. 2019 [4] stresses that the mechanism of Amazon deforestation is the product of the traditional form of continuous expansion of the agricultural frontier in Brazil, with the occupation of (private or public) virgin lands, the (il) legal extraction of timber, the introduction of extensive livestock farming[8] and, subsequently, the development of a more modern

---

5   In the beginning of the 2000s, it was around 25 million hectares. That drop represented a substantial improvement, caused mainly by very strong command and control policies.

6   Foley, J.A. et al. (2007) [10] shows this in a very clear way.

7   Moutinho et al. (2016:2) observes [8] that: "A vast body of literature discusses the principal drivers of deforestation in the Brazilian Amazon (Nepstad et al., 2001; Kaimowitz et al., 2004; Fearnside, 2005; Etter et al., 2006; Scouvart et al., 2008; Boucher et al., 2011; Guerra, 2014; Nepstad et al., 2014; Azevedo-Ramos et al., 2015). There is still, however, no consensus concerning which intervention was the most effective in prompting the dramatic reduction in deforestation in the region since 2005."

8   Reydon (2011) [3] shows that the main driver of the transformation to livestock farming is, on one hand, the large amounts of vacant land to be grabbed, linked to the possibility of introducing livestock at low cost, rendering deforestation an unbeatable capital appreciation strategy. A survey conducted by the National Institute for Space Research (INPE) showed that 62.2 % of the near 720,000 km² clearance of forest, was occupied by pastureland.

agriculture and livestock sector[9]. These economic activities exercise the role of generating income, legitimizing short-term occupation by new squatters, virtually without the need of many resources[10]. In the long run, the land remains with more intensive livestock farming or, if there is demand, it will be converted to grain farming or other agricultural activities.

However, what is important for occupation or deforestation is the existence of an expectation that there will be demand for this land[11], to be used at some point in the future, causing its price to rise significantly. The closer it is to being used productively, the higher the land value appreciation.

The macro policies as the turnarounds and the changing governments in Brazil also influenced the deforestation levels. After 2014, the federal government's macro policies changed toward economic austerity-oriented policies. In 2016, there was a major turmoil with the impeachment[12] that resulted in institutional instability and the deposition of Dilma Rousseff. From 2016 onward, the austerity pattern on macro policies became more intense and in relation to the deforestation, also coupled with a conservative push against social and environmental policies, the result is a weakened institutional capacity, especially through cuts in environmental, social, and science-related governmental branches [13,14]. After Bolsonaro's election in 2018 with an anti-environmental, anti-indigenous people and pro-deforestation rhetoric, the area deforested in the Amazon increased again, as shown by Figure 1. Based on scattered information, the increase in deforestation in 2020 is still higher, and it is happening in unowned land and in indigenous reservations, which is also caused by mining activities.

## 3. Undefinition of Land Rights and Deforestation: Some Evidence

To diminish deforestation in the Amazon biome, besides the more general policies[13] that must impact the Amazonian region as a whole, there is a need to fine tune the policies associated with the land ownership and responsibilities. Therefore, it is important to have an overview of what kind of properties are deforesting in order to establish the effective policies.

The only existing information are estimations based on satellite images. As can be seen in Table 1 adapted from Moutinho et al.(2016) [8], most deforestation in 2016 happened on private properties, then rural settlements with 35.5% and 28.7%, respectively. To diminish the deforestation on this type of land, the main policies in place are the Forest Code and other specific policies such as law 13,465, which will not be discussed here. Between 2012 to 2015, the summed deforestation of the categories 'land with no information', 'federal', and 'state lands' was always around 37% of the total deforestation. This is a typical kind of land over which there is no control, as the federal and state governments do not have clear cadasters of their land. The reason for the fall of this participation in 2016 to 25.1% is still unknown, but it might be that all private landholders that registered at CAR started to be private properties. Furthermore, what is important to highlight from this table is that private landowners deforested more in 2016, reaching 2462 km$^2$, 35.5% of the total. It is expected that all of the deforestation on private properties is legal, but that is still not possible to confirm, as Forest Code (2012) authorizes only 20% of private properties to be deforested for productive use in this region.

---

9    Reydon et al. (2019) [4] also argues that speculation with land in general and the conversion of forest into pastureland are important drivers of the deforestation.

10   It is frequently these same occupiers who make use of slave labor.

11   This is the result of increases in the prices of beef, soy, or even of reports that Brazil is going to be the largest alcohol producer in the world. In recent times, these factors have converged, causing the demand for land to grow even more as well as its price, encouraging deforestation.

12   Or coup, depending on one's point of view.

13   Most studies agree that command and control policies played an important role in the diminishing of the deforestation at the beginning of the 2000s.

**Table 1.** Deforested area in the Amazon by land title category from 2012 to 2016. Adapted with permission from [8] Moutinho et al. (2016).

| Agrarian Category | 2012 | | 2013 | | 2014 | | 2015 | | 2016 | |
|---|---|---|---|---|---|---|---|---|---|---|
| | (Km²) | (%) | (Km²) | (%) | (Km²) | (%) | (Km²) | (%) | (Km²) | (%) |
| Indigenous Lands | 168 | 3.8 | 170 | 3.2 | 71 | 1.6 | 62 | 1.2 | 88 | 1.3 |
| Federal Conservation Unit | 175 | 4 | 187 | 3.5 | 120 | 2.8 | 184 | 3.5 | 201 | 2.9 |
| State Conservation Unit | 117 | 2.7 | 175 | 3.3 | 174 | 4 | 233 | 4.4 | 322 | 4.6 |
| Environmental Protection Areas | 124 | 2.8 | 228 | 4.3 | 202 | 4.6 | 245 | 4.7 | 207 | 3.0 |
| Rural Settlements | 1239 | 28.3 | 1518 | 28.7 | 1269 | 29.2 | 1437 | 27.3 | 1986 | 28.6 |
| Private properties | 986 | 22.5 | 1009 | 0 | 883 | 20.3 | 1113 | 21.2 | 2462 | 35.5 |
| Federal Public Lands | 574 | 13.1 | 743 | 14.1 | 584 | 13.4 | 670 | 12.7 | 855 | 12.3 |
| State Public Lands | 15 | 0.3 | 31 | 0.6 | 0 | 0 | 7 | 0.1 | 59 | 0.9 |
| No information | 982 | 22.4 | 1222 | 23.1 | 1047 | 24.1 | 1306 | 24.8 | 758 | 10.9 |
| Grand total | 4381 | 100.0 | 5282 | 100.0 | 5350 | 100.0 | 5256 | 100.0 | 6938 | 100.0 |

The other important information from Table 1 is that the smallest amount of deforestation happened on indigenous land, and in all kinds of conservation units and protected areas. It is clear that the main effort to maintain the Amazon forest is related to clarifying property rights: giving out titles to private owners, establishing clear boundaries for indigenous territories, protected areas, among other types of land use, but also to have a good, mapped cadaster of it all to enforce the Forest Code and its protection rules.

It is insufficient to know what kind of land has been deforested to avoid further deforestation; there is also a need to understand the amount of land that is under risk for each of these types. One study by Azevedo-Ramos, and Moutinho [15] stated that: "what is not widely known is that 70 million hectares ha of that public land—an area nearly twice the size of Germany—remains undesignated." (p. 125). They estimated the amount of land that is yet to be designated and that there should be a specific policy for its protection and avoidance against its deforestation. However, the ownership, possession, or responsibility over those 70 million ha were not clearly defined by the authors. In Figure 2, it can be seen that these areas are mostly in rather accessible areas where deforestation can happen easily. However, this definition of public forests is not very precise, as it comes from the cadaster of the *Serviço Florestal Brasileiro*. Mostly, there are people in those areas and there is a need to know what the real agrarian situation is, and if that land is under risk of deforestation.

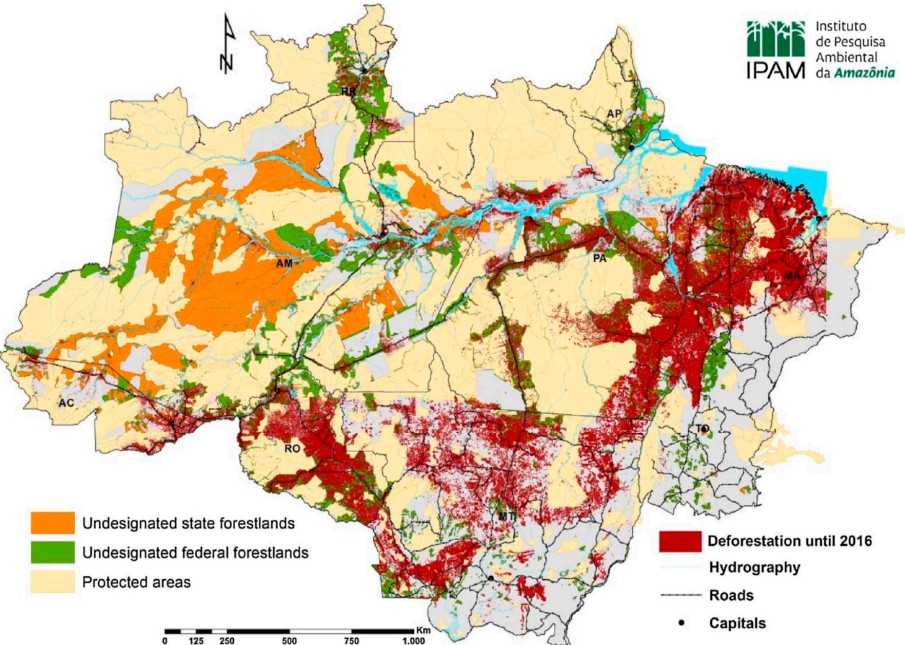

**Figure 2.** Public forests of the Brazilian Legal Amazon and its 70 million hectares of undesignated forestlands [15].

Sparoveck et al. [5] were able to estimate the different kinds of properties existing in the total area of Brazil using other sources and satellite images, as demonstrated by Table 2. This table integrates all information existing in the country and makes it possible to start understanding the real agrarian situation of Brazil. One important result that has come out of this is that 196,056 million ha have no clear allocation as they are undesignated or unregistered land[14].

**Table 2.** Area and number of units of Brazilian land tenure categories [5].

| Land Tenure Category | Area (ha) | % | Number | % |
|---|---|---|---|---|
| Indigenous Reserves | 112,412,239 | 13.2% | 600 | 0.0% |
| Conservation Unit[15] | 93,403,026 | 11.0% | 1337 | 0.0% |
| Community Territory | 1,779,373 | 0.2% | 815 | 0.0% |
| Military | 3,006,965 | 0.4% | 104 | 0.0% |
| Rural Settlement | 41,736,096 | 4.9% | 7547 | 0.2% |
| Undesignated Lands | 54,599,607 | 6.4% | 22,016 | 0.5% |
| *Total Public Land* | 306,937,306 | 36.1% | 32,419 | 1% |
| Private property from CAR[16] | | | | |
| Small | 83,400,520 | 9.8% | 3,805,698 | 79.0% |
| Medium | 42,077,338 | 4.9% | 167,537 | 3.5% |
| Large | 48,366,589 | 5.7% | 34,779 | 0.7% |
| Private property from SIGEF[17] | | | | |
| Small | 12,700,175 | 1.5% | 206,070 | 4.3% |
| Medium | 41,551,394 | 4.9% | 110,830 | 2.3% |
| Large | 134,531,227 | 15.8% | 62,677 | 1.3% |
| Private property from Terra Legal Program | 9,830,630 | 1.2% | 116,854 | 2.4% |
| Quilombola Territory | 3,117,971 | 0.4% | 378 | 0.0% |
| *Total Private Land* | 375,575,843 | 44.2% | 4,504,823 | 94% |
| Unregistered land | 141,454,569 | 16.6% | | |
| Transportation network, Urban area and Water bodies | 26,310,500 | 3.1% | 280,692 | 5.8% |
| Total Brazil | 850,278,218 | 100.0% | 4,817,934 | 100% |

To have an idea of its location, the best way is to look at Figure 3, which demonstrates that most of these areas are in the Amazon region. Thus, what can be concluded from the previous information is that there is a strong need to clarify land ownership, legitimize occupants, and build a good land administration system to enforce those limits. This combination could protect the forest and make deforestation much more difficult.

To understand the large amounts of undesignated land, Reydon et al, (2019) [4] showed that land with no information or is public land plays an important role because since the Land Law of 1850, it has been legally defined that whatever land that is not private and registered at the registration offices is State land. By doing so, Brazilians developed a 'habit' of grabbing this kind of land and later obtaining documentation for it (by any means necessary) because it was easier and possible to do so. In the article by Reydon at al. (2019) [4], this phenomenon also helps to explain how the Brazilian Land Administration

---

[14]  Public lands that have not been designated to a final use. The findings of Sparoveck et al. (2019) [5] differ from the 65.5 million ha of undesignated forest lands in the Amazon found by Azevedo-Ramos and Moutinho (2018) [15] due to the hierarchy rules adopted, where forest type B have a low level of priority and are classified as other categories. Forests type B include Federal or State lands covered with forests whose final designations have not been decided yet. They are under the administration of the Brazilian Forest Service (SFB). Additionally, Sparoveck et al.'s (2019) [5] estimations are relative to the whole country, but surely most of the undesignated and unregistered land are in the Amazon region.

[15]  We excluded APAs from the conservation unit category. APA (area of environmental protection) is a type of conservation unit of sustainable use which may occur in areas of public or private domain that allow human occupation and economic activities including intensive agriculture. Its creation does not imply expropriation of private land ownership. It sums 44 million ha. Its inclusion would confuse interpretation of land ownership and overlaps as it necessarily coincides with other land tenure categories.

[16]  Cadastro Ambiental Rural (Rural Environmental Registry).

[17]  Sistema de Gestão Fundiária—INCRA (Land tenure management system from INCRA).

System was built and why its malfunctioning is at the core of many Brazilian problems, especially that of deforestation.

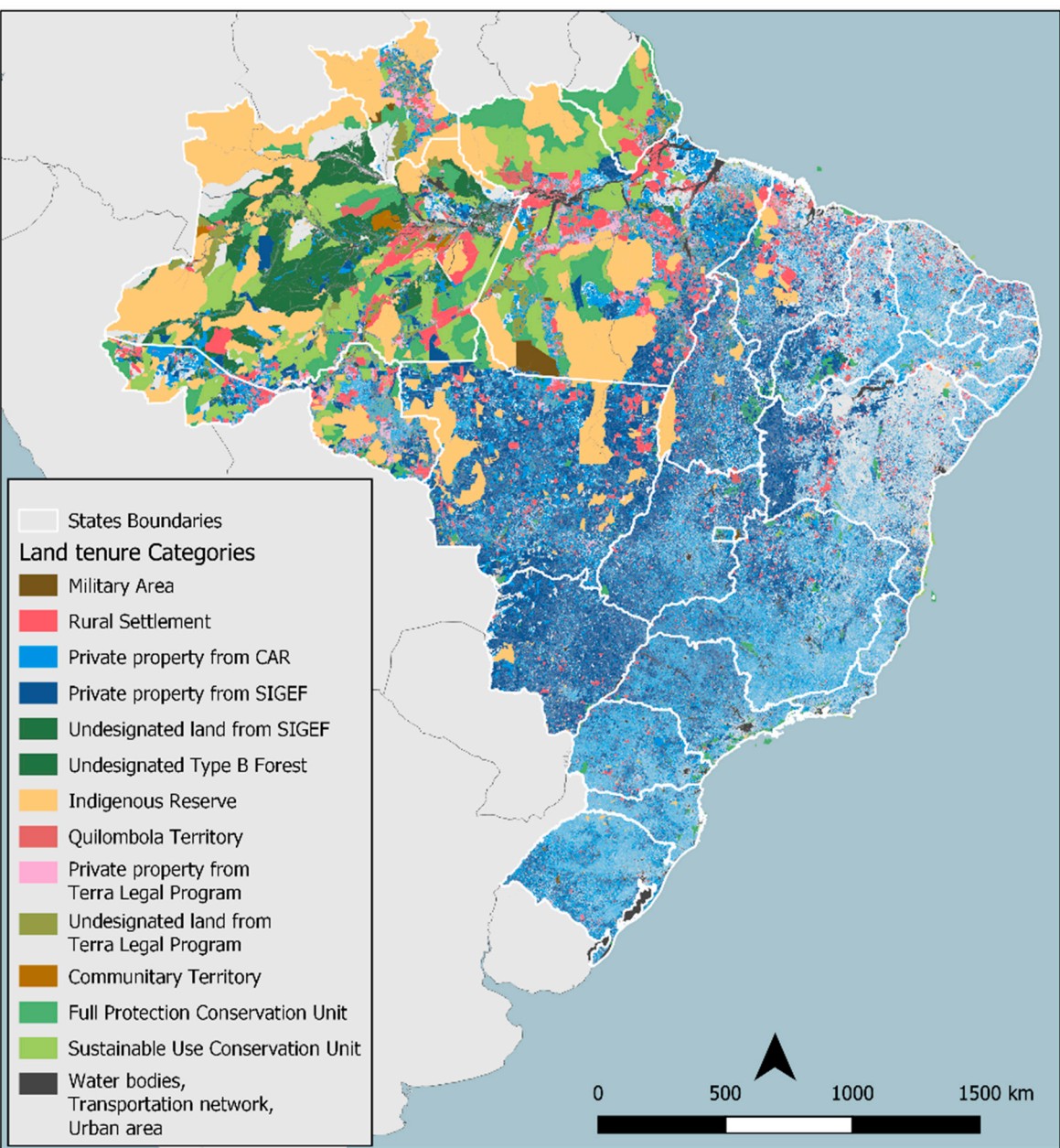

**Figure 3.** Reprinted with permission from Sparoveck et al. (2019) [16] from the Atlas of Brazilian Agriculture.

It is also important to highlight that as the system does not work properly, the main efforts have been directed to create land ownership cadasters. In the same article, Reydon et al. (2019) [4] explained CNIR as the official mapped landownership cadaster from INCRA and *Receita Federal*, which has SIGEF as its operational system, and that CAR was created by the Environmental Ministry with the Forest Code and is a self-mapped cadaster of land users.

This is the reason that the issue of land registration, as per the legislation discussed above, is so important in the country. If the registration of possessions is possible, with large amounts of land 'available' due to its legal uncertainty, this will act as an incentive for land grabbers to officialize land holdings.

The experience of Terra Legal, a federal government program to clear ownership on registered national public land, achieved important results between 2009 and 2017 [17],

which was based on mapping, the titling of small landholders, and transferring larger amounts of land to indigenous and environmental reservations. As this was done, part of the public land was registered at the registration offices, so there would be no future controversy around it [18]. However, as it was shown, the amount of land that is still unregistered is still very large, and those areas are suffering larger amounts of pressure for occupation, conflicts, and deforestation. Therefore, the clearing of ownership rights in these areas is very important to avoid further similar issues.

## 4. Why Good Land Administration Reduces Deforestation

In Reydon et al. [4], it was possible to show that the lack of a good land administration system in Brazil is an immense problem and plays an important role in deforestation.

When analyzing the Brazilian Amazon rainforest, Araujo et al. (2009) [19] found that insecure property rights had a positive impact on deforestation in the period of 1988–2000 and, therefore, guaranteeing clear and secure rights on rural land ownership could decrease or avoid future deforestation.

In the same way, Assunção et al. (2015) [20] calculated the avoided deforestation in the same region through coordinated public policies between various levels of government and showed that in the period of 2004–2009, about 59% of the predicted deforestation was avoided for this reason.

Finally, a meta-analysis study, made with more than 118 published articles, by Robinson et al. (2014) [9] contributes to the conclusion that land tenure security is significantly associated with lower rates of deforestation, adding that this occurs regardless of the specific land access regime (tenure, property, customary systems, etc.). What this study shows, based on international literature, is that the clarification of land rights is very important as the main instrument to stop deforestation.

In Brown et al. (2016) [21], through a different approach, the authors analyzed the effects of land occupation on deforestation in Brazil, reaching the conclusion that in an environment of the low security of property titles, and with policies that value land deforestation over forested land, occupation has a direct influence on deforestation including occupations in a given municipality that can affect deforestation in adjacent areas.

More than one analyzed study deals with the forms of governance of forests and indigenous communities and their impacts on deforestation. Blackman et al. (2017) [22] analyzed one of these cases in Peru and concluded that the titling of these communities reduced deforestation by two thirds in a period of two years after the program.

Fernandes (2018) [23] showed, with systematic review methodology[18], that improvements in land governance had an observable positive impact on economic development, more specifically on the economic aspects of (a) production, productivity, and access to credit; (b) diminishing of poverty; and (c) dynamization of land markets. However, it also has large impacts on women's rights over land and on environment protection (deforestation and erosion decrease).

In Reydon et al. [4] (pp. 12), the authors concluded that:

"It is clear that, in order to combat deforestation in the Brazilian Amazon, it is necessary to move forward with improvements in the Land Administration/Governance System. The land cadaster must be completed and integrated with the CAR. The regularization of ownership, along the lines of the Terra Legal program, must be continued and expanded to include state-controlled public lands. Once a reliable land cadaster is in place, land taxes can be improved, which in turn diminish speculation and improve control over forested lands and associated environmental crimes. Finally, it is important to stress that, while better land governance is a necessary precondition for reducing deforestation, it is not in itself sufficient. Land governance and environmental protection must be structured under long term compromises and insulated against the politics of the day. "

---

18 Systematic review methodology consists of: "[...] studies which synthesize [sic] all the existing high-quality evidence using transparent methods to give the best possible, generalizable statements about what is known" (Waddington et al. 2012, p. 360) [24].

In conclusion, Reydon et al. [4] proposed that what is needed to diminish rainforest deforestation, mainly in the Amazon, but also in other regions, is:

a.　　an improvement in the land administration/governance system, mostly the integration of cadaster and registry systems (CNIR, SIGEF, SINTER, CAR, and others), so that every landholder can be identified and localized[19].

b.　　The regulation of ownership similar to Terra Legal has to be continued and amplified to public land that are under national and states' responsibility.

Without secure property rights, farmers cannot obtain access to investment loans or public benefits and are almost invisible to the government. Without land administration reflecting the realities on the ground, land governance is difficult: the government cannot promote sustainable planning, which happens mostly at the cost of the natural environment and vulnerable groups. The main existing argument is that the areas are too big and with the existing technology, it is not possible to identify all landholders' rights in the Amazon region. The next section presents a case in Brazil where the small landholder's regulation process was quick, affordable, scalable, and successful. Another example of a case in Colombia is also mentioned, as it used the FfP method to clarify land rights and avoid conflicts to enhance the possibility of forest preservation.

## 5. Materials and Methods

### 5.1. Method—A Participatory Fit-For-Purpose (FfP) Approach in Areas under Deforestation Pressure

Forests are frequently located in different arrangements of property rights from communal rights to private and public ones, among others; so it needs a strong intervention to solve eventual controversies and recognize all those rights in a legitimate way. To achieve this, this article will show how to identify and cadaster landholders using the participatory fit-for-purpose methodology, which was very clearly defined by Ennemark et al. [25] (p. 6) as:

"the approach used for building land administration systems in less developed countries should be flexible and focused on citizens' needs such as providing security of tenure and control of land use, rather than focusing on top-end technical solutions and high accuracy surveys." A fit-for-purpose approach includes the following elements:

- Flexibility in the spatial data capture approaches to provide for varying use and occupation.
- Inclusive in scope to cover all tenure and all land.
- Participatory in approach to data capture and use to ensure community support.
- Affordable for the government to establish and operate, and for society to use.
- Reliable in terms of information that is authoritative and up-to-date.
- Attainable in relation to establishing the system within a short timeframe and within available resources.
- Upgradeable with regard to incremental upgrades and improvement over time in response to social and legal needs and emerging economic opportunities.

A country's legal and institutional framework must be revised to apply the elements of the fit-for-purpose approach. This means that the fit-for-purpose approach must be enshrined in law, it must still be implemented within a robust land governance framework, and the information must be made accessible to all users.

Considering this, the FfP method should be quick, affordable, and as accurate as possible so that it can solve concrete land ownership conflicts. To do so, the methodology must be flexible to fit different institutional settings and the technical demands of each location. These determinations must be respected in real scenarios, but its results will enable discussion on the viability of those standards and settings. Furthermore, the methodology must consider all different types of rights and legitimate occupants, therefore, the participation of all neighboring parties are central to it. Due to the involvement of all parties, any conflict

---

[19]　For more information on the existing Brazilian Cadasters and Registering System, see Reydon et al. (2019) [4] and Reydon et al. (2017) [17].

resolution is much easier, once the final data are plotted for all participants to see and understand the nature of the overlap or if there are any illegitimate claims. This is why an affordable technology, with good-enough accuracy, can provide much good and spur development, especially in regions with weak or unclear land rights.

An example of the use of FfP methodology with good results that can be an inspiration for the issues in the Amazon region was conducted in Santa Teresita del Tuparro, an indigenous protected area located in Cumaribo, Vichada, Colombia[20]. It has been constituted as a special reserve since 1983, with an area of 180,000 ha, but most of its boundaries are determined by natural boundaries such as rivers, or dirt roads. More critically in the southern area of the reserve, there was an imminent conflict regarding overlapping claims.

Due to this factor, the pressure over the land and the risk of conflicts increased largely and for that, a FfP intervention was necessary. Therefore, the parcels in the current cadaster were used to identify the conflict and the actors in the area. Even though it did not show the actual reality, it was used to understand the dispute between the Santa Teresita del Tuparro indigenous groups and the adjacent parcels of farmers (colonos). The mapping of the conflicting claims was done using a FfP approach to clear the conflict by understanding the origin of the overlap.

The problem relied on the titles given in the past by the state and their poorly defined boundaries, which made the local parties understand that the land dispute was not being caused by them. The dispute could be solved by showing the indigenous people and the farmers the maps that where self-measured by them, which made them confident of the results and the official cadaster that was being used.

The fact that there were real and accurate data on the perceived limits helped to show the exact part of the land that the dispute and what the problem was about. After this, their rights were clearer and the communities were engaged in solving their overlapping boundaries in a peaceful way by recognizing their rights among their peers and neighbors, with the certainty that these areas will be respected after this process.

As shown before, land rights play an important role to reduce deforestation in the long run, especially in undesignated land in the Amazon region. In this section, a case study that applied this methodology will be detailed to illustrate and reinforce the potential of this method to solve complex land issues associated with deforestation in a participatory way. The next case shows a successful experience that will contribute to similar situations in the Amazon region. The case was the 'Tangará da Serra' in Brazil, demonstrating that it is possible to clarify property rights and provide formal titling to small land holders in a quick and low-cost way. It helps to show how it is possible to assure different land rights and peacefully solve conflicts over land using the FfP participatory approach.

The costs and timeframe necessary to regulate these situations are also central aspects of this methodology. Due to this, it is important to find experiences that would allow estimations of real costs and the minimum timeframe necessary, but also to extrapolate the results and conclusions to a national perspective. Once it is understood or has identified land holders, communal land, state lands, private possessions, or any other arrangement, it is necessary to estimate the feasibility of the current legislation, procedures/regulations, and expansive accuracy standards, especially considering the goal of having all land holdings identified and mapped within an updated national cadastral system. Not only for legal reasons, but the correct definition of all kinds of land rights, from community based to legitimate possession, are necessary for communities to thrive.

The importance of the formal recognition of land rights and the institutional capacity to enforce them has already been discussed, but there are few innovative ways to solve such complex issues. For example, in Brazil, a very controversial law at the time (Law 13,467 of 2017, the "Land Regularization Law") was enacted that, among some widely criticized aspects, eased the administrative regulation process that empowered registry offices to regularize properties in situations where there were no conflicts or disputes over land or

---

20    See Molendijk et al. (2020) [26] for a complete description of the case.

boundaries in a much simpler configuration than the usual judicial process. The main innovation presented in the new legislation is the need of a formal agreement between neighbors for their shared boundaries that must be registered within the property deed (including the size, shape, and borders of the property). This formal agreement on the boundaries between neighbors gives the registry office enough security to go through the process with the certainty that property rights are being formalized with consent, thus speeding up the process without compromising any of the parties. This was also one of the reasons why a case in Brazil was chosen in order to evaluate the gains and consequences of this legal change.

### 5.2. Materials and Equipment's

To implement this theoretical approach, different test cases were carried out using GNSS receptors (Global Navigation Satellite System) from Trimble®R1 and R2, which are usually simpler than those used to georeference properties; in exchange, it has been used as an accessible and affordable technology, with adequate accuracy for mapping rural plots. The usage of this simple technology by the local population can be seen in Figure 4.

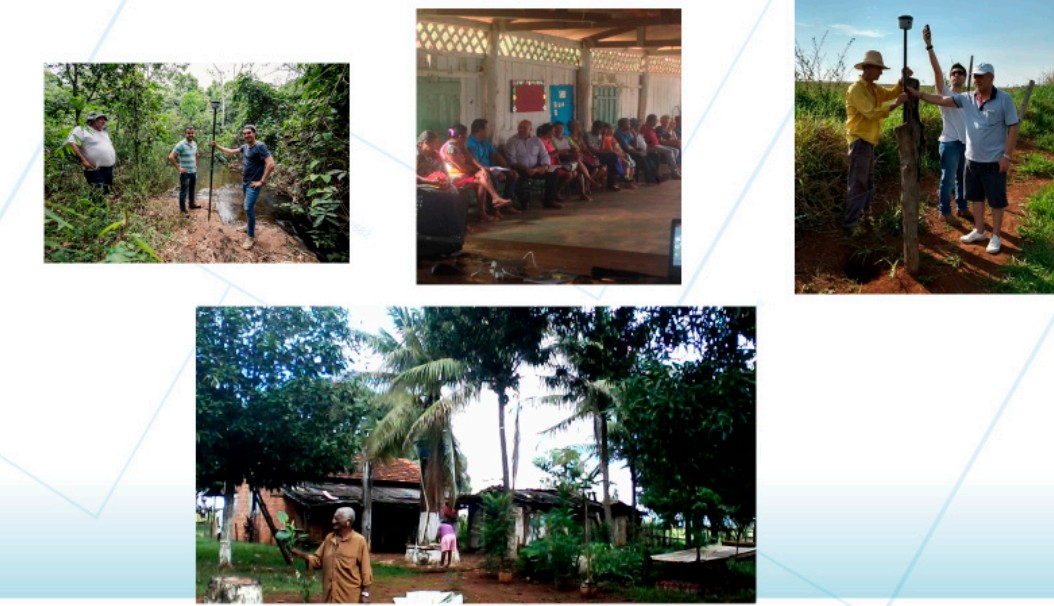

**Figure 4.** Images from the pilot case in Tangará da Serra.

Different countries and regions have different legal and technical demands for recognizing rights over land including different legal settings that must also be respected. For example, INCRA in Brazil, which is the national regulator of accuracy standards regarding these matters, requires a maximum error for georeferencing a property of 0.5 m to be certified. Other regulations may vary the level of precision, but the main argument brought by the FfP methodology is that many rights and conflicts could be solved by using a less precise survey[21], but still 'good enough' to speed up the regulation process, otherwise it may take decades and millions of dollars to do so.

Considering the Brazilian context, it is also important to highlight Law no. 10,267 enacted in 2001, which determines, among other relevant things, that all private properties should be certified by the INCRA under these accuracy standards. The same law sets a time limit for rural property owners to seek certification based on the size of their properties: the largest properties have a stricter timeframe, which has long been overdue, and smaller properties (up to 100 hectares) are supposed to expire by 2023. The compulsory certification

---

[21] Considering the Brazilian legal standards, a minimum horizontal accuracy of 0.5 m is required for 'artificial limits', three meters for 'natural limits', and up to 7.5 m in 'inaccessible limits'.

processes, determined by INCRA itself, demands an expensive georeferencing/mapping of the property because it must be done by a verified and contracted engineer with specific precision standards even in difficult terrain. Considering this institutional framework, a fit-for-purpose approach becomes extremely necessary, so it is better to be pro-active and start collecting evidence from the field of the advantages of a FfP methodology than to push forward this public issue regarding informality for years to come. Surely as soon as the properties are defined, the owners will have to comply with the Forest Code, therefore, maintaining 80% of their areas covered with native vegetation.

## 6. The Tangará Case: From Georeferencing to Titling under Six Months

As shown for the Brazilian Amazon, property rights are fragile or absent. This not only leads to serious conflicts over land and increases deforestation, but also hampers economic growth. Evidence shows that informal rights outnumber formal land rights in Brazil, for both urban and rural areas. Formalizing land rights can be very time consuming (over 20 years for a conflict resolution over a parcel within the judicial system is not an exception) and costly (around R$ 30.000 or USD 9.138[22])[23], which makes it a challenging task, especially for smallholder communities that are social and economically vulnerable.

To address these challenges of land regularization, a participatory FfP approach was developed in Tangará da Serra, in the state of Mato Grosso, Brazil during the year 2018. The initial goal of this pilot project was to develop a test case for the implementation of a fit-for-purpose methodology adapted to Brazil's legal and institutional framework, which could contribute to a viable model for a country-wide regularization process. Furthermore, an area in the Amazon region was chosen to test that reality. Therefore, the project was carried out by the Dutch Cadaster and Land Registry (Kadaster) and the State University of Campinas (UNICAMP), in collaboration with INCRA (the responsible national institution for rural land administration, cadaster, and certification), the local registry office ("cartório"), a local lawyer to represent the case, and a responsible engineer to certify the final maps.

The project consisted of a FFP approach on a small rural community with a selection of farmers, who could not afford to go through the usual process individually, starting with the social mapping of smallholder's properties that did not have any legal documents that officially stated their rights over their specific piece of land. This lack of formality is due to the time spent and financial burden that is required to seek regularization, and therefore, most of their parcels remained unregistered and unmapped.

The original focus of the project was to test a method for georeferencing properties (compliant with the accuracy standards demanded by the INCRA) as a viable solution for mapping tenure in adverse possession for smallholders in Brazil. By using a much more affordable technology and a social mapping strategy with the community including public inspection where neighbors sign to agree on the location of their boundaries, the intention was to optimize the timeframe and reduce costs at the maximum. As will be shown, this experience was very useful for all kinds of identification and land holding clarification.

First, Tangará da Serra is an interesting case because all types of land tenure and land use can be found there: commercial farms, small farmers, informal tenure, indigenous lands, natural areas, and state land, and all of them could be addressed in this pilot. Second, it is situated in the Amazon region where most land related problems are such as the overlapping of rights, the absence of clear rights, the invasion of public land, and others. Third, as all of the institutional stakeholders in Tangará da Serra had a keen interest in participating in this project, it was very feasible to prove the advantages of the FFP method with this arrangement. Especially considering the institutional complexity and limitations of the land registry as one of the top bottlenecks of securing land rights in Brazil, since in

---

[22]  Using an approximate exchange rate (from 15 June 2017) of 1 USD being 3,283 BR$, according to the Brazilian Central Bank (https://www.bcb.gov.br/conversao).

[23]  Reydon (2010) [27] estimated the cost for georeferencing the whole country based on the regularizing experience at the municipality of São José das Pontas in the state of Pará.

Tangará da Serra they were partners, a strong case could be made in an integrated approach to improve land administration in the region.

In Figure 5, the mosaic of properties that are available on the cadaster from the state of Mato Grosso and the municipality of Tangará da Serra is visible. It offers a scattered view on the existing land situations and people relations: a discontinuous map of tenure, showing 'islands' of formalized properties within an uncertain background. In Mato Grosso, as in Brazil as a whole, there is still the need for a national complete georeferenced cadastral map (with continuous coverage of the whole country instead of the current 'patchwork' map of land tenure), registered, and integrated with all of these smaller plots and also those that still have not obtained formal recognition of their rights.

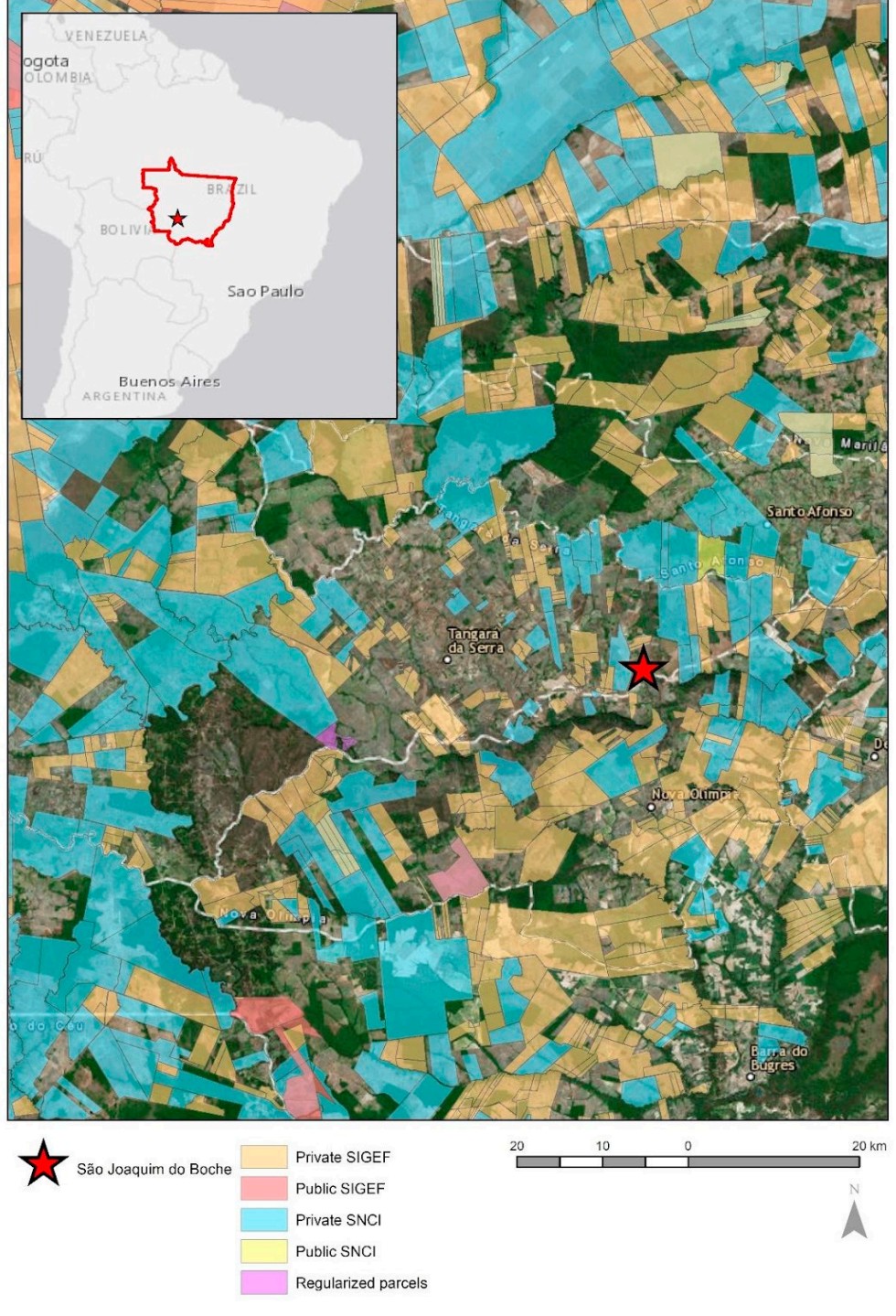

**Figure 5.** Land tenure mosaic of the Tangará da Serra region and intervention case location.

Tangará da Serra provides us with a unique opportunity of work in Mato Grosso and in Brazil due to its pioneer initiative of mapping all known properties that are already registered and by other secondary information from Brazil's official cadaster system that are within the municipality. Therefore, we were able to identify an unregistered community that could be formalized by the fit-for-purpose methodology. Then, the community of "São Joaquim do Boche", in the rural area of the municipality was chosen to be part of this initiative, with 60 known parcels that include formal georeferenced properties, informal georeferenced plots, and a vast majority of informal, not georeferenced plots. From those, we excluded the already registered plots and used the georeferenced ones for a precision survey comparison between thee FFP methodology and an official survey that had been done by a regular technician.

This test case started at the end of 2017 and within four months of coordinated work, 52 rural properties were regularized in January 2018 (with no costs to the smallholders, while complying with the current legal standards of the state of Mato Grosso and Brazil), as it can be seen on Figure 6. Although we managed to reduce the legal costs in the project to the minimum, there were still many costly procedures that are officially required during the regularization process such as the georeferencing by a verified engineer, the legal assistance provided by a lawyer, and the costs regarding the registry office practices. Nevertheless, the intentions on reducing costs allowed us to understand one important bottleneck for a national wide regularization, especially for smallholders, where many will not be able to pay for the process individually. As the Brazilian state is currently facing an enormous deficit, the public service is also not likely to cover these costs. Therefore, there is an urgent need to look for FfP solutions, leading to a fast, affordable, and complete land administration.

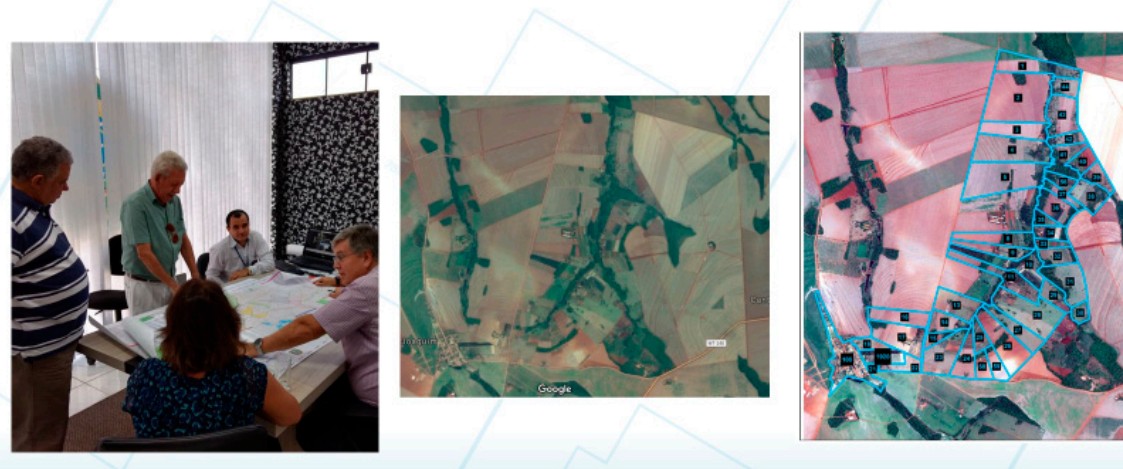

**Figure 6.** Images of the fit-for-purpose experience in Tangará da Serra.

The technical results obtained from this experience presented an average of error (using the R2 device, the most sophisticated one) of 0.31 meters, which is below the required standard. Furthermore, INCRA's normative requires a certified engineer to design the georeferenced map of the property (which is very costly), although an interpretation of the law allowed us to prepare the maps using the FfP approach and a hired engineer was only required to validate the maps, optimizing his work and the cost–benefit relation for the process of the regularization of smallholders. For private landowners, besides providing the legal security rights, it proved that social mapping helps smallholders to have considerably more secure tenure rights, which prevent conflicts.

The results from this project have already led to significant changes in national policies and made policy makers aware of the problem and complexity of the current requirements.

Very much inspired by the technical results obtained by the FfP social mapping approach, an update in the National Cadastral System (SIGEF) was proposed and named as "SIGEF 2.0." The new proposed system will integrate the social mapping of adverse possessions of land use, confronting the obtained information with the already certified land properties (public and private) and (if not in conflict) certifying informal rural households. Through this, the INCRA will promote a "good-enough" tenure on land, securing land holders that are in areas where there are no conflicts of rights with any other claims in a complete mapping of the current situation.

## 7. Conclusions

As the burning and deforestation of the Brazilian Amazon forest and other rainforests in the world play such important roles in the global climate equilibrium and on the emission of greenhouse gases, its control plays a very important role in the SDG's agenda.

This article started showing that the deforestation in the Brazilian Amazon started growing again in 2019 and that it occurred mostly on public land or undesignated land. Based on recent studies, it was possible to show that these areas sum up to near 200 million ha, about 25% of the Brazilian surface. Not all of it is in the Amazon region or with forest cover, but in the Amazon region, this is the type of land that is mostly possible to be grabbed, deforested, and used for speculative reasons.

Furthermore, the article presented evidence, based on international literature, that clear property rights are essential to the preservation of primary forests all around the globe. The article concluded by showing one participatory land rights clarification case in the Amazon region using fit-for-purpose methodology to help forest preservation. From the study, it is expected to mainstream this methodology to help to clarify the ownership of small landholders, the rights of traditional population landholders as well as diminish potential conflicts over undesignated public land. It also aimed to find ways to improve the legislation and the institutional settings to make the clarification of property rights easier and thus ultimately help maintain the Amazon rainforest.

**Author Contributions:** Conceptualization, B.R. and M.M.; methodology, B.R., M.M. and G.S.; software, N.P.; investigation, N.P. and G.S.; data curation, B.R., M.M., N.P. and G.S.; writing—original draft preparation, B.R.; writing—review and editing, B.R. and G.S.; project administration, B.R. and M.M.; supervision, B.R. and M.M.; funding acquisition, M.M. All authors have read and agreed to the published version of the manuscript.

**Funding:** The funding of the publication costs for this article has been kindly provided by the School of Land Administration Studies, University of Twente, in combination with Kadaster International from the Netherlands.

**Institutional Review Board Statement:** Not applicable.

**Informed Consent Statement:** Not applicable.

**Data Availability Statement:** Not applicable.

**Acknowledgments:** Acknowledge the contribution from the registrar José de Arimatéia for making this project viable and the good will of the people from Tangará da Serra.

**Conflicts of Interest:** The authors declare no conflict of interest. The funders had no role in the design of the study; in the collection, analyses, or interpretation of data; in the writing of the manuscript, or in the decision to publish the results.

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
