# Peer review of "The Amazon Forest Preservation by Clarifying Property Rights and Potential Conflicts: How Experiments Using Fit-for-Purpose Can Help"

_land, doi:10.3390/land10020225_

Round 1

Reviewer 1 Report

The manuscript can potentially make a valuable contribution to the literature on deforestation in the Brazilian Amazon. There are two main potential contributions. The first is clarification about the quantity and type of unregistered and undesignated land in the region and the presentation of evidence that a significant amount of deforestation occurs in these areas (although even more occurs on private lands and rural settlements). The second contribution is the description of a small-scale exercise of Fit-for-Purpose land regularization which could potentially be scaled up to regularize and designate the unregistered and undesignated lands.

The paper could be improved in three ways:

1) by emphasizing the link of deforestation and unregistered and undesignated lands ideally with more evidence. It would also strengthen the paper to explain why it is not focusing on deforestation on private land or rural settlements which are subject to monitoring and enforcement through the CAR and Forest Code. It would benefit from clarifying the scope of the manuscript in this regard while taking into account findings about deforestation on land registered in the CAR in the literature such as found in:

Alix-Garcia, J., L. Rausch, J. L’Roe, H. K. Gibbs, and J. Munger. 2017. Avoided deforestation linked to environmental registration in the Brazilian Amazon. Conservation Letters.

L’Roe, J., L. Rausch, J. Munger, H.K. Gibbs. 2016. Policy intersections, land registration, and forest clearing patterns in the Amazonian state of Para. Land Use Policy 57: 193-203

2) by providing a more explicit discussion of the existing and proposed processes for land regularization, forest designation and forest protection (CNIR, SIGEF, SINTER, CAR and others) in terms of both their opportunities and limitations; and

3) a more explicit account of how the pilot  FfP exercise can influence these processes and how it may or may not be relevant for wider scale land designations like new protected areas or state forests and areas beyond the immediate agricultural frontier. 

The section 5.1 on the FfP experience in Colombia has interesting details on the FfP process there, but the foreign case is peripheral to the scope of the manuscript and leads away from the central thread of the argument. It would appear to be more in line with the aim of the manuscript to focus section 5 on the  existing processes for registering and regularizing land in the Brazilian Amazon, so that the unique contribution of the FfP process discussed in 5.2. could be understood in context. 

The manuscript would benefit for editing for clarity and style. 

Author Response

1rst Reviewer

- by emphasizing the link of deforestation and unregistered and undesignated lands ideally with more evidence –> More references were added in that sense

- explain why it is not focusing on deforestation on private land or rural settlements -> “As most studies on deforestation deal with the private properties, the main aim of this article is to emphasize that, besides the efforts of reducing deforestation on private properties, there is an urgent need to clarify land rights in general, but more intensively on public and undesignated land.” “One important information that comes out of it is that 196,056 million ha have no clear allocation as they are undesignated or unregistered land[1]. So, those are the areas that are most under pressure for occupation, conflicts, and deforestation. Therefore, the clearing of ownership rights in these areas is very important to avoid new conflict.”

- more explicit discussion of the existing and proposed processes for land regularization, forest designation and forest protection -> In line 177 there are two footnotes for explaining the Terra Legal program and why it is a reference for how land regularization in the amazon should be done.

- how the pilot  FfP exercise can influence these processes and how it may or may not be relevant for wider scale land designations -> “, two successful experiences, although not comparable, may contribute to similar situations in the Amazon region, being the first case of ‘Tangará da Serra’ in Brazil, demonstrating that it is possible to clarify property rights and provide formal titling to small land holders in a quick and low cost way, and the second one in Colombia, helps to show how it is possible to assure different land rights and peacefully solve conflict over land using the FfP participatory approach.”.

[1] Public lands that have not been designated to a final use. Sparoveck et al. (2019) [7] findings differ from the 65.5 million ha of undesignated forest lands in the Amazon found by Azevedo-Ramos & Moutinho (2018) [14]. due to the hierarchy rules adopted, where Forests type B have a low level of priority and are classified as other categories. Forests Type B are Federal or States lands covered with forests which final designations have not been decided yet. They are under the administration of the Brazilian Forest Service (SFB). Besides that, the Sparoveck et al. (2019) [7] estimations are relative to the whole country. But surely most of the undesignated and unregistered land are in the Amazon region.

Reviewer 2 Report

The Amazon forest preservation by clearing property rights and potential conflicts: how experiments using FFP can help

I have reviewed the manuscript with great interest in the topic at hand, but I have found many formal defects and I also have serious doubts about the scientific ethics of this work which is copied from an existing one on the Internet.

  • FFP appears in the title as in the abstract, but it is not defined anywhere, and I honestly don't know what it refers to.
  • The references in the manuscript are very confusing. It is cited both with the format of the journal, numerical order, and with footnotes.
  • The references in the references section do not follow any format. Reference 14 is the most indicative of how badly it is referenced: [14]Blackman et al (2017),
  • Decimal separators use commas instead of points. e.g. line 40 “205,8 million hectares. 205,8 million hectares. These represents 24,2 %”

In table 1 the decimal point or comma is used interchangeably as a separator.

  • In table 2, the comma appears to be a separator of thousands.
  • The objective of the work is not clear
  • Figure 1, cites a source TerraBrasilis (inpe.br), cuyos datos no coinciden con los de la figura 1. g. E.g. In 2019 it was 9762 km2, but according to the source it was 10129 km2.

  • Figure 2 is a literal copy of the article published by Azevedo-Ramos, C., & Moutinho, P. (2018). No man’s land in the Brazilian Amazon: Could undesignated public forests slow Amazon deforestation?. Land use policy73, 125-127.

This is not acceptable.

  • Table 1 claims to be based on the reference 7. Azevedo-Ramos, C., & Moutinho, P. (2018). No man’s land in the Brazilian Amazon: Could undesignated public forests slow Amazon deforestation?. Land use policy73, 125-127.

But this work is only two pages long, and this data is not in that work. Or I can't find it. The authors should clarify this aspect.

Checking reference 1, this information does appear in the reference 1. REYDON, B. P., FERNANDES, V. B., & TELLES, T. S. (2019). Land governance as a precondition for  decreasing deforestation in the Brazilian Amazon. Land Use Policy, 88. 104313.

In short, the references do not seem to be as rigorous as required for a scientific work.

  • Figure 3, is from a web page (which is referenced), Atlas Agropecuário (imaflora.org). But it is not clear what the authors mean by this image. An original one should be made or this should be clarified.

  • Line 184 In [1] (pp. 8-9) the authors concluded that: “Land governance will not solve the problem of deforestation in Amazonia, …. “ leida la referencia REYDON, B. P., FERNANDES, V. B., & TELLES, T. S. (2019). Land governance as a precondition for  decreasing deforestation in the Brazilian Amazon. Land Use Policy, 88. 104313. This text cannot be found after reading this reference.
  • In reference to [1] Lines 199 to 201. cadaster and registry systems (CNIR, SIGEF, SINTER, CAR and others). Reference 1 has no citation or reference to SINTER.

  • line 214 methodology that [16] (p.6) was proposed as: From lines 215 to 234 are copied form that reference. This must be avoided.

  • 1. The case of Cumaribo in Colombia 235. As authors claim “This part is very much based on Molendijk, M; Reydon, B (2020), report from project Land and Peace in Colombia”,. I am not sure if this is acceptable.

  • Checking this link

https://www.fig.net/resources/proceedings/fig_proceedings/fig2020/papers/ts06h/TS06H_reydon_molendijk_et_al_10614.pdf

                One can see that this is this manuscript. Including the figures, where they keep the same numbers of the original figures.

The manuscript has too many formal and scientific errors to be accepted for publication. It must be rejected

Author Response

2nd Reviewer

  • FFP appears in the title as in the abstract, but it is not defined anywhere, and I honestly don't know what it refers to. -> Defined by Ennemark in line 262
  • The references in the manuscript are very confusing. It is cited both with the format of the journal, numerical order, and with footnotes. -> see if the changes helped. Some are in footnotes because of the length of the article
  • The references in the references section do not follow any format. Reference 14 is the most indicative of how badly it is referenced: [14]Blackman et al (2017), -> All references were fixed. Please double-check.
  • Decimal separators use commas instead of points. e.g. line 40 “205,8 million hectares. 205,8 million hectares. These represents 24,2 %” In table 2, the comma appears to be a separator of thousands. -> Fixed all over the text.
  • The objective of the work is not clear -> As most studies on deforestation deal with the private properties, the main aim of this article is to emphasize that, besides the efforts of reducing deforestation on private properties, there is an urgent need to clarify land rights in general, but more intensively on public and undesignated land.
  • Figure 1, cites a source TerraBrasilis (inpe.br), cuyos datos no coinciden con los de la figura 1. g. E.g. In 2019 it was 9762 km2, but according to the source it was 10129 km2. -> updated figure was put in place.
  • Figure 2 is a literal copy of the article published by Azevedo-Ramos, C., & Moutinho, P. (2018). No man’s land in the Brazilian Amazon: Could undesignated public forests slow Amazon deforestation?. Land use policy73, 125-127. -> and for that it is referenced
  • Figure 3, is from a web page (which is referenced), Atlas Agropecuário (imaflora.org). But it is not clear what the authors mean by this image. An original one should be made or this should be clarified. -> we changed the picture to an English version of it. See if it helps.
  • Line 184 In [1] (pp. 8-9) the authors concluded that: “Land governance will not solve the problem of deforestation in Amazonia, …. “ leida la referencia REYDON, B. P., FERNANDES, V. B., & TELLES, T. S. (2019). Land governance as a precondition for  decreasing deforestation in the Brazilian Amazon. Land Use Policy, 88. 104313. -> this quote was fixed. Please check the lines 218-234
  • In reference to [1] Lines 199 to 201. cadaster and registry systems (CNIR, SIGEF, SINTER, CAR and others). Reference 1 has no citation or reference to SINTER. -> we poited in footnotes where the reader can find further information, but there is not enought room to explain it all
  •  
  • line 214 methodology that [16] (p.6) was proposed as: From lines 215 to 234 are copied form that reference. This must be avoided. -> It is arranged and referenced as such.
  •  
  • The case of Cumaribo in Colombia 235. As authors claim “This part is very much based on Molendijk, M; Reydon, B (2020), report from project Land and Peace in Colombia”,. I am not sure if this is acceptable. -> that note was excluded.
  •  
  • Checking this link   

    https://www.fig.net/resources/proceedings/fig_proceedings/fig2020/papers/ts06h/TS06H_reydon_molendijk_et_al_10614.pdf

    One can see that this is this manuscript. Including the figures, where they keep the same numbers of the original figures. ->

    An much earlier version of this paper was approved by the FIG 2020 conference. Besides that it was not published anywhere else. Conferences procedures are not considered publication. Regarding the copyright issue with an article presented at the FIG 2020, we understand it as a previous version to discuss the experience at the conference, with the intent to collect critics and ideas for this current version submitted to LAND. We recognized your concern and during the revision and reconstruction of the article a lot of it has changed, now this article is much less involved in the technical aspects of the field work survey and much more centered to FfP methodology as a possible solution for clarifying land rights, meaning that it can be published at the LAND.

Reviewer 3 Report

This paper is a relevant contribution to the land governance debate, as it presents how participatory land referencing experiences have enabled to clarify land rights. Literature does indeed show that lack of land rights is a driver of deforestation and the authors summarize relevant data from Brazil that converge with this theory, as deforestation in public and undesignated land is highest. They then present how two interesting experiences which used the Fit for Purpose methodology have enabled to support smallholders to define the limits of their properties and achieve regularization.

My main concern with this paper is that the authors do not present the different voices in this land governance debate and seem to consider that clarifying land through georeferencing is a technical matter and easy to solve with simplified methods. However, this is a highly politicized debate in Brazil (“medida provisoria da grilhagem” (MP 910/2019) and “projeto de lei da grilhagem” (2.633/2020)) and I am unconfortable with the fact that the authors, who are certainly very aware of this debate, don’t clearly mention the limits to the technical approach. Indeed, it is urgent to regularize land for smallholders in Brazil, to guaranty their rights, but most entities who support smallholders and traditional populations in Brazil have been against the measures and law projects supported by the federal government these past 4 years, as they underline that auto-declaration opens the way to land grabbing. The authors should at least mention this debate (I have found several scientific references presenting it, see for example Kluck 2020 in the Revista da ANPEG), present the risks involved in simplified land regularization, and show how their experiences have tried to deal with these risks. Authors could also better qualify the interdependencies between land and environmental policies, in particular between SIGEF and CAR, what are the limits of each policy  and the importance of integrating them.

Another strong limit in the current version is that although many interesting elements are presented in this paper, it is difficult to see how they connect within a common narrative. The general organization is messy, attesting to this are the repeated number parts. It is not clear what the authors consider methodology and results, and the status of each experience (Columbian and Brazilian) is confusing: is Columbia a methodological reference, then applied to Brazil? Is the intention of the authors to compare both experiences? This choice must be clear and will lead to different configurations in the article. If the focus is on Brazil (as it seems since the literature review only mentions Brazil), present Columbia very briefly in a methods’ section, to show how FFP has been used previously and then show how it was applied in Brazil, the challenges in this context, its success and limits. If the intention is to compare, you should follow the same presentation for each case and draw common lessons. Either way, I would have liked to find more details on the process, if it was part of a project, which institutions were involved, how the farmers participated in the process, what the different stakeholders learned from this, how it "inspired" larger policy (line 465).

Summing up, I believe this work deserves being published but I recommend the authors to focus on presenting the experiences, developing more the process and potential risks and safeguards.

Punctual remarks:

l 2: careful! Clearing is used for forest clearing, here you mean “clarifying property rights” (in general, you need a good English proof, there are many errors and typos)  

l 4: write in extenso FFP as it is not straight forward for most readers

l 29: by using “slash and burn” it seems like smallholders, who are the main farmers to use this practice, are the main responsible actors for GHG. Slash-and-burn are only a small part of fires in the Amazon region, most emissions come from large scale fires. For this, see for example Carmenta et al. 2011 in Ecology and Society.

l 33: instead of fotenote, give reference in text

l 36: quoting is not done in the appropriate form, all through the article. If you say “by… “ cite the author and then put the number between brackets. See example in other articles of Land.

l 40, footnote 2: when citing references in footnote, you must also list the reference at the end of the article. This reference does appear as [8], but latter on, which is confusing. In general, you should cite within the text and not so much in footnotes.

l 58: I do not feel that section 2 is fundamental for this article, go directly to deforestation and property rights and summarize essential information from part 2

l 108-109: “deforestation that occurred until 2015 mostly happened where land rights were not clearly established, that is: on lands with no information, federal and state land” => this sentence is not accurate, correct “until 2015” by “from 2012 to 2016; there is hardly any deforestation on state public land; deforestation on public land is moderately high, but lower than on private land; most deforestation is in rural settlements, which you do not present (use studies, for example from IPAM, to specify this type of deforestation, which could reinforce your argument regarding uncertain land rights).

l 113: indeed, with this table, you can’t prove that deforestation in private properties is legal or not, but it is evident that deforestation is high in this category: this could turn the arguments against you: once farmers own their property, they may obtain authorization to legally deforest. This institutional subversion was evidenced by Rajão et al, 2012, in Public Administration and Development. Duchelle et al 2014 in World Development also mention this contradiction. You must better inform this debate.

l 135: figure 2, in the undesignated land figure rural settlements, which are a special case but don’t appear on this map. There is a better map of IPAM (from 2017) which evidences rural settlements.

l 145: for readers who are not familiar with Brazil land policy, explain what is Terra Legal

l 153: “that combination is the only way forward to protect the forest”… this sound very normative and not so appropriate in a scientific paper, either find references to back this up or reformulate

l 155: I do not understand what this figure brings more than figure 2 (and is very difficult to read)

l 184: since you are the ones who say this, reformulate and bring out what is most important quoting this size of text is not necessary

l 208: description of FFP should come as method: who designed the methodology, better explain why it is important to be participatory, situate with relation to other similar participatory land governance methodologies

l 235: fotenote 20: reference does not appear at the end, cite in text and give complete reference at the end of the paper

l 236: this case is about an indigenous territory, not undestinated land, which is a very different case regarding deforestation (as you showed, in Brazil, indigenous land have very low deforestation figures), this makes it unclear how the case fits in your narrative

l 245: when did this take place? Who were the partners? Who were the collectors? How was the method discussed with local people?

l 313: what is ITC?

l 320: what method was used to avoid conflict and promote an agreement?

l 328: Figure 5, put legend in English

l 339: clarified instead of clearer; “happily engaged” does not sound very scientific; we would like to know more about the outcome, if it really is a case study (and not just a methodological reference)

l 350: how was the value in dollars calculated? Give a reference year. In 2010, change between real and dollar was 1.76, which would give $15,000. Currently, change is much higher, and would be around $5600.

l 362: somewhere, present the community, how families arrived (land settlement, informal possession?), how many families there are today

l 396: Is Tangara “unique”, or is it representative? I’m not sure “unique” is the best way to qualify this project and it seems to me, as you showed in the initial parts, that it is quite representative of a common reality in the Brazilian Amazon

l 400: who chose Tangara? Who chose São Joaquim do Boche? The governance of the project is not clear.

l 407: do you have an estimation of the cost of one regularization for the project? Consideration on cost are interesting, but they could come in a “lessons” section, after presenting the process

l 415: and who could pay for FFP? NGOs?? Sub-contracting between state administration and private technical services?

l 459: you ought to enter in more details on how the forest policy can be (or not) better implemented once land rights are clarified, in particular linking with the Rural Environmental Cadaster (CAR) which was barely mentioned in the paper. Although it is probably not the place to detail the Brazilian Forest Code (which since 2012 should be more correctly named as a law), I wonder if the 80% mentioned here are accurate, since Tangara da Serra is situated both in the Amazon and Cerrado biome. You could at least present the average deforestation of properties, as most of the satellite images seem to show properties with less than 80% forest cover, which seems to suggest that farmers will only be held to maintain the forest they have.  

l 461: figure 8 or figure 6? Check all captions. Moreover, it would have been important, apart from presenting equipment, to present the social process involved, how farmers were chosen, how they were involved, the conflicts that might have arisen?

l 465: “very much inspired”: explain this up-scaling process or how experience/lessons were integrated into the National System: technical notes or presentation by researchers?

l 485: “the article concluded by showing two participatory land rights clarification cases”: are they only as conclusion? I find that they should be the center of the article

Author Response

3rd Reviewer

  • is that the authors do not present the different voices in this land governance debate and seem to consider that clarifying land through georeferencing is a technical matter and easy to solve with simplified methods. However, this is a highly politicized debate in Brazil (“medida provisoria da grilhagem” (MP 910/2019) and “projeto de lei da grilhagem” (2.633/2020)) and I am unconfortable with the fact that the authors, who are certainly very aware of this debate, don’t clearly mention the limits to the technical approach -> Although we do find this debate very important and pertinent, the article won’t have enough room for a proper discussion. We did add a footnote to guide the readers that do want to acknowledge this debate.
  • The authors should at least mention this debate (I have found several scientific references presenting it, see for example Kluck 2020 in the Revista da ANPEG), present the risks involved in simplified land regularization, and show how their experiences have tried to deal with these risks. -> As recommended, we added Kuck and other references to sustain the debate and arguments.
  • It is not clear what the authors consider methodology and results, and the status of each experience -> we changed the structure of the article to make it clearer. Please check if the new configuration with ‘Material and Methods’ separated from the ‘Cases’ helped.
  • l 2: careful! Clearing is used for forest clearing, here you mean “clarifying property rights” (in general, you need a good English proof, there are many errors and typos)  -> Sorry for that! We tried to correct all of those mistakes and typos in a more watchful review,.
  • l 4: write in extenso FFP as it is not straight forward for most readers -> also considered and corrected through out the text.
  • l 29: by using “slash and burn” it seems like smallholders, who are the main farmers to use this practice, are the main responsible actors for GHG. Slash-and-burn are only a small part of fires in the Amazon region, most emissions come from large scale fires. For this, see for example Carmenta et al. 2011 in Ecology and Society. -> as recommended, we changed the phrasing
  • l 33: instead of fotenote, give reference in text -> some of those we did it, but for others we preferred footnotes as the article was already too long
  • l 36: quoting is not done in the appropriate form, all through the article. If you say “by… “ cite the author and then put the number between brackets. See example in other articles of Land. -> we corrected all quotes and references accordingly
  • l 40, footnote 2: when citing references in footnote, you must also list the reference at the end of the article. This reference does appear as [8], but latter on, which is confusing. In general, you should cite within the text and not so much in footnotes. -> all refences within footnotes were included in the reference list
  • l 58: I do not feel that section 2 is fundamental for this article, go directly to deforestation and property rights and summarize essential information from part 2 -> we changed the structure of the article to make it more direct to the point
  • l 108-109: “deforestation that occurred until 2015 mostly happened where land rights were not clearly established, that is: on lands with no information, federal and state land” => this sentence is not accurate, correct “until 2015” by “from 2012 to 2016; there is hardly any deforestation on state public land; deforestation on public land is moderately high, but lower than on private land; most deforestation is in rural settlements, which you do not present (use studies, for example from IPAM, to specify this type of deforestation, which could reinforce your argument regarding uncertain land rights).-> thank you for your clear observation. The sentence was corrected
  • l 113: indeed, with this table, you can’t prove that deforestation in private properties is legal or not, but it is evident that deforestation is high in this category: this could turn the arguments against you: once farmers own their property, they may obtain authorization to legally deforest. This institutional subversion was evidenced by Rajão et al, 2012, in Public Administration and Development. Duchelle et al 2014 in World Development also mention this contradiction. You must better inform this debate. -> we changed the form of argument and the structure of the article. Please check if the changes made it clearer.
  • l 135: figure 2, in the undesignated land figure rural settlements, which are a special case but don’t appear on this map. There is a better map of IPAM (from 2017) which evidences rural settlements. -> we chose this image for the categories as presented, including protecte3d areas, and also the accumulated deforestation, as explained by the reference quoted.
  • l 145: for readers who are not familiar with Brazil land policy, explain what is Terra Legal -> we added two footnotes and references for the readers to better understand Terra Legal Program. Please, bear in mind that there is not enough room in the article for a detailed explanation.
  • l 153: “that combination is the only way forward to protect the forest”… this sound very normative and not so appropriate in a scientific paper, either find references to back this up or reformulate -> agreed upon. We changed phrasing to avoid this statement.
  • l 155: I do not understand what this figure brings more than figure 2 (and is very difficult to read) -> we changed figures to a better one. Lines 181-185 try to explain the figure better.
  • l 184: since you are the ones who say this, reformulate and bring out what is most important quoting this size of text is not necessary -> we changed quotes to make it better.
  • l 208: description of FFP should come as method: who designed the methodology, better explain why it is important to be participatory, situate with relation to other similar participatory land governance methodologies -> that was one of the reasons that justified a complete change in design and structure of the article. Please check if the new format helped.
  • l 235: fotenote 20: reference does not appear at the end, cite in text and give complete reference at the end of the paper -> all references and footnotes were corrected. Sorry for the trivial mistake.
  • l 236: this case is about an indigenous territory, not undestinated land, which is a very different case regarding deforestation (as you showed, in Brazil, indigenous land have very low deforestation figures), this makes it unclear how the case fits in your narrative -> The main contribution from Cumaribo’s experience was a peaceful arrangement in a conflicted area, demonstrating that a participatory approach can ease conflict in the most difficult situations. See if the changes made helped to highlight this message.
  • l 245: when did this take place? Who were the partners? Who were the collectors? How was the method discussed with local people? -> Because Cumaribo was not a central point to the article, we made it shorter and not all information could be included. We could add more details about it but the maximum length of the article will not allow it.
  • l 313: what is ITC? -> removed after revision
  • l 320: what method was used to avoid conflict and promote an agreement? -> we tried to explain the participatory approach of FFP in the Section ‘Methods’. By this, we expect that some questions about it were solved.
  • l 328: Figure 5, put legend in English -> all legends of Figures are now in English. What was not translated is the name of the reserve or is in italic.
  • l 339: clarified instead of clearer; “happily engaged” does not sound very scientific; we would like to know more about the outcome, if it really is a case study (and not just a methodological reference) -> We changed the case studies content to make them more relevant to the arguments of the article.
  • l 350: how was the value in dollars calculated? Give a reference year. In 2010, change between real and dollar was 1.76, which would give $15,000. Currently, change is much higher, and would be around $5600. -> we added a footnote for exchange value at the time of the project was being conducted, accordingly to the Brazilian Central Bank.
  • l 362: somewhere, present the community, how families arrived (land settlement, informal possession?), how many families there are today -> We could add more details about all cases but the maximum length of the article would not allow it.
  • l 396: Is Tangara “unique”, or is it representative? I’m not sure “unique” is the best way to qualify this project and it seems to me, as you showed in the initial parts, that it is quite representative of a common reality in the Brazilian Amazon -> After the reconfiguration of the article’s structure, we tried to explain ‘why’ Tangará as an introduction of the sub-chapter – lines 355-400
  • l 400: who chose Tangara? Who chose São Joaquim do Boche? The governance of the project is not clear. -> After the reconfiguration of the article’s structure, we tried to explain ‘why’ Tangará as an introduction of the sub-chapter – lines 355-400
  • l 407: do you have an estimation of the cost of one regularization for the project? Consideration on cost are interesting, but they could come in a “lessons” section, after presenting the process -> in this specific case there were no costs to the local population, but estimates were made according to the current legislation.
  • l 415: and who could pay for FFP? NGOs?? Sub-contracting between state administration and private technical services? -> Anyone of those agents given as example, there is no such limitation for FFP, that is why we did not comment on it.
  • l 459: you ought to enter in more details on how the forest policy can be (or not) better implemented once land rights are clarified, in particular linking with the Rural Environmental Cadaster (CAR) which was barely mentioned in the paper. Although it is probably not the place to detail the Brazilian Forest Code (which since 2012 should be more correctly named as a law), I wonder if the 80% mentioned here are accurate, since Tangara da Serra is situated both in the Amazon and Cerrado biome. You could at least present the average deforestation of properties, as most of the satellite images seem to show properties with less than 80% forest cover, which seems to suggest that farmers will only be held to maintain the forest they have.  -> after correction, we added more information about the CAR influence on private parties and how regularization might help to enforce such compliance. Please check lines 346-351 and 139-147. Although important, the CAR was not the main focus of our discussion and the limit in length prevented us from taking a deeper dive into it.
  • l 461: figure 8 or figure 6? Check all captions. Moreover, it would have been important, apart from presenting equipment, to present the social process involved, how farmers were chosen, how they were involved, the conflicts that might have arisen? -> All figures were corrected, sorry for the trivial mistake. Please bear in mind the limit in length that won’t allow too much detail for the cases.
  • l 465: “very much inspired”: explain this up-scaling process or how experience/lessons were integrated into the National System: technical notes or presentation by researchers? -> With the new configuration of the article, we tried to make it clearer the possibilities of scaling up the solutions presented, but no concrete changes were made afterwards.
  • l 485: “the article concluded by showing two participatory land rights clarification cases”: are they only as conclusion? I find that they should be the centre of the article -> because they are central to the article, they are the part of the conclusion. Much of the results are present along the article to make it shorter, not only summarized in the conclusion.

Round 2

Reviewer 1 Report

This paper has benefitted from some revisions, and the focus on public lands is more clearly explained in the revised manuscript. However the revised manuscript still provides almost no explanation of how the Brazilian land administration actually is intended to work in the undesignated/unregistered public areas in question, with the result that the key argument of the paper--that improved land administration on unregistered and undesignated parcels of public land will reduce deforestation--is weakened. A series of institutions and programs for land administration (CAR, SIGEF, SINTER, Terra Legal) are introduced in both text and tables without explaining what these are, how they are supposed to work, how they are actually working and how the case study of FfP land administration improves on the operation of these systems. The Law on Regularization (Law 13,467 of 2017) is introduced on line 301 but without any explanation of how it applies to the apparently large unregistered/undesignated areas. Something like flow chart depicting how public land designation, registration of ownership rights, and registration of parcels in the Rural Environmental Registry is supposed to operate in forested areas could help address this gap. It would also be valuable to understand how the FfP approach described in the first case study improves on or solves the problems associated with this system. The Colombia case study continues to seem out of place in the manuscript focused on Brazil. 

Author Response

However the revised manuscript still provides almost no explanation of how the Brazilian land administration actually is intended to work in the undesignated/unregistered public areas in question, with the result that the key argument of the paper--that improved land administration on unregistered and undesignated parcels of public land will reduce deforestation--is weakened.

More information regarding the limitations of the Brazilian Land Administration System was added.

 A series of institutions and programs for land administration (CAR, SIGEF, SINTER, Terra Legal) are introduced in both text and tables without explaining what these are, how they are supposed to work, how they are actually working and how the case study of FfP land administration improves on the operation of these systems.

More information on these Systems were given, but also, directions for where to find further information was also pointed out, since there is not enough room in the article to disclose them in detail.

The Law on Regularization (Law 13,467 of 2017) is introduced on line 301 but without any explanation of how it applies to the apparently large unregistered/undesignated areas. Something like flow chart depicting how public land designation, registration of ownership rights, and registration of parcels in the Rural Environmental Registry is supposed to operate in forested areas could help address this gap. It would also be valuable to understand how the FfP approach described in the first case study improves on or solves the problems associated with this system. The Colombia case study continues to seem out of place in the manuscript focused on Brazil. 

 More information on the general procedures regarding public land destination were added, but in order to propose a specific chart for it, there would be the need to explain the figure and its details, where there is no longer space in the article for it.

The Colombian case was excluded

Reviewer 2 Report

The authors have satisfactorily resolved all the doubts and suggestions posed. For me the version can be accepted.

Minor Comments

The references still do not follow the style of the Journal. See the authors' guide.

Author Response

The English was improved by a native speaker. 

Reviewer 3 Report

Article has improved but authors must still work on clarity of some sentences.

1/ Bring in references to back-up your affirmations

Among places where more references are needed:

L32: for example INPE data : With the Bolsonaro administration there was an evident escalation of the deforestation in Brazil, which grew the discussion around it.

L42: In literature, there is a perception that deforestation occurs traditionally when land is grabbed or bought to be used immediately or in the long run.

L112: The closer it is to being used productively, the higher the land value appreciation.

L304: among some widely criticized aspects

2/ Some elements are missing

L328: can be solved by using a less precise survey => it would be important to give us an idea of what is the resolution so we can understand how different it is from INCRA standard

L437: INCORA=> what is it ?

I understand you don’t have much space, but it’s fundamental to indicate the year in which were carried out the FFP processes, so the reader knows when to situate the case:

L  353  To address these challenges of land regularization, a participatory FfP approach was developed in Tangará da Serra, state of Mato Grosso, Brazil.

L 445: A first meeting was held with leaders of around 30 communities from the resguardo Santa Teresita del Tuparro. => what year?

3/ Comment on answers that still need to be improved in text:

l 108-109: “deforestation that occurred until 2015 mostly happened where land rights were not clearly established, that is: on lands with no information, federal and state land” => this sentence is not accurate, correct “until 2015” by “from 2012 to 2016; there is hardly any deforestation on state public land; deforestation on public land is moderately high, but lower than on private land; most deforestation is in rural settlements, which you do not present (use studies, for example from IPAM, to specify this type of deforestation, which could reinforce your argument regarding uncertain land rights).-> thank you for your clear observation. The sentence was corrected

Reviewer: Date was not corrected (l 130). Moreover, I still see no explanation to the fact that most deforestation is in rural settlements: since the figure 2 considers rural settlements in “undesignated land”, you could comment on this regarding the previous table.

and correct: it is not correct that there is much deforestation in state public land, on the contrary, it is quite low (l 132)

l 145: for readers who are not familiar with Brazil land policy, explain what is Terra Legal -> we added two footnotes and references for the readers to better understand Terra Legal Program. Please, bear in mind that there is not enough room in the article for a detailed explanation

Reviewer: I understand that you don’t have much space but you don’t even say what is Terra Legal, just say it’s a federal program for land regularization

l 184: since you are the ones who say this, reformulate and bring out what is most important quoting this size of text is not necessary -> we changed quotes to make it better.

Reviewer: I think lines 232 to 236 are sufficient, no need for total quote

4/ There are still many English errors in the manuscript.

You need to have your text revised by a native English-speaker. For example, just in the abstract:

"The burning and the deforestation of the Brazilian Amazon forest that has been  highlighted recently and occurs mostly on public or undesignated land." => several problems in this sentence

"proposes based on concrete case of participatory clarification of land rights in forest regions using Fit for Purpose (FfP) methodology, helps forest preservation." => problem in this sentence

Author Response

As the Colombian case was excluded the comments scratched are due to dat.

1/ Bring in references to back-up your affirmations

Among places where more references are needed:

L32: for example INPE data : With the Bolsonaro administration there was an evident escalation of the deforestation in Brazil, which grew the discussion around it.

L42: In literature, there is a perception that deforestation occurs traditionally when land is grabbed or bought to be used immediately or in the long run.

L112: The closer it is to being used productively, the higher the land value appreciation.

L304: among some widely criticized aspects

More references and information’s were added to back-up these affirmations, as for the others, they were excluded during the reformulation of the article. regarding the limitations of the Brazilian Land Administration System was added 

2/ Some elements are missing

L328: can be solved by using a less precise survey => it would be important to give us an idea of what is the resolution so we can understand how different it is from INCRA standard => A footnote was added

L437: INCORA=> what is it ?

L  353  To address these challenges of land regularization, a participatory FfP approach was developed in Tangará da Serra, state of Mato Grosso, Brazil. I understand you don’t have much space, but it’s fundamental to indicate the year in which were carried out the FFP processes, so the reader knows when to situate the case:

=> A specific date was added

L 445: A first meeting was held with leaders of around 30 communities from the resguardo Santa Teresita del Tuparro. => what year?

3/ Comment on answers that still need to be improved in text:

l 108-109: “deforestation that occurred until 2015 mostly happened where land rights were not clearly established, that is: on lands with no information, federal and state land” => this sentence is not accurate, correct “until 2015” by “from 2012 to 2016; there is hardly any deforestation on state public land; deforestation on public land is moderately high, but lower than on private land; most deforestation is in rural settlements, which you do not present (use studies, for example from IPAM, to specify this type of deforestation, which could reinforce your argument regarding uncertain land rights).-> thank you for your clear observation. The sentence was corrected

Reviewer: Date was not corrected (l 130). Moreover, I still see no explanation to the fact that most deforestation is in rural settlements: since the figure 2 considers rural settlements in “undesignated land”, you could comment on this regarding the previous table.

and correct: it is not correct that there is much deforestation in state public land, on the contrary, it is quite low (l 132)

This comment led to major revisions in text. Clearly there was a misinterpretation of the data and we adapted the text to make it more clear on this statement. Please verify if the revision covered your legitimate concern.

l 145: for readers who are not familiar with Brazil land policy, explain what is Terra Legal -> we added two footnotes and references for the readers to better understand Terra Legal Program. Please, bear in mind that there is not enough room in the article for a detailed explanation

Reviewer: I understand that you don’t have much space but you don’t even say what is Terra Legal, just say it’s a federal program for land regularization

More information on the Terra Legal program, its goals and main results were also added. Please verify if now it is suitable.  

l 184: since you are the ones who say this, reformulate and bring out what is most important quoting this size of text is not necessary -> we changed quotes to make it better.

Reviewer: I think lines 232 to 236 are sufficient, no need for total quote

We reviewed the need to keep the entire quote considering your comments. Because of that, a significant part of it was excluded. As for the rest, we still believe it brings important contributions to the text and the general argument.  

4/ There are still many English errors in the manuscript.

You need to have your text revised by a native English-speaker. For example, just in the abstract:

"The burning and the deforestation of the Brazilian Amazon forest that has been  highlighted recently and occurs mostly on public or undesignated land." => several problems in this sentence

"proposes based on concrete case of participatory clarification of land rights in forest regions using Fit for Purpose (FfP) methodology, helps forest preservation." => problem in this sentence

We truly believe that this observation may have been done in a previous version sent, as these statements where already corrected after the English revision by a native speak. Please verify the new version submitted now and see if there is still need for any further revision of that nature.

Round 3

Reviewer 1 Report

This manuscript has been substantially improved in revision and clarifications have been made to address previous reviewer comments. A thorough editing for English usage and style will improve its clarity. 

Author Response

We are sorry for the inconvenience with the minor English mistakes, the professional reviewer was contracted before these last changes and some details may have passed by our attention.

Reviewer 3 Report

It find the paper much more clear without the Colombian case, I believe this was a good choice. However, check the text to be coherent with this:

L70: "but also will show concrete examples of actions" => a concrete example

L77: "Cases using" => case / L79: "two case studies" => change

L262: in Columbia => suppress sentence

Most of my comments were considered. However, some still need to be better informed:

L137: I still do not understand why you add 'state land" here, as there is hardly any deforestation on state land

L365: footnote 24 => I wanted to know what was the accuracy of your survey, not what was tolerated by the law - I then found the accuracy L456, answering the initial consideration

Although English was much improved, you still need to have a final review. For example:

L29, suppress "as it": The burning and deforestation of the Brazilian Amazon forest which has been recently highlighted byinternational press, (as it) plays an important role on the global climate equilibrium and on the global Greenhouse Gas (GHG) emissions, an important aspect of the Sustainable Development Goals (SDG).

L55, suppress (it): With the Forest Code, (it) was created the CAR => give the meaning of CAR and function (not only to monitor deforestation, but map existing land use)

L86: 69to 11.1km2 => 6.9 and misses thousand!!!

Author Response

It find the paper much more clear without the Colombian case, I believe this was a good choice. However, check the text to be coherent with this:

L70: "but also will show concrete examples of actions" => a concrete example - check

L77: "Cases using" => case / L79: "two case studies" => change - check

L262: in Columbia => suppress sentence – we are also introducing the mention for the Cumaribo case in L 304 to 309.

Most of my comments were considered. However, some still need to be better informed:

L137: I still do not understand why you add 'state land" here, as there is hardly any deforestation on state land – we would very much like to keep the mention that State, as federal, do not have a proper cadastre. Even if the deforestation problem related to state lands is much smaller than the rest. But in prewies years the deforestation there was higther. 

L365: footnote 24 => I wanted to know what was the accuracy of your survey, not what was tolerated by the law - I then found the accuracy L456, answering the initial consideration – So we suppressed ok?

Although English was much improved, you still need to have a final review. For example:

L29, suppress "as it": The burning and deforestation of the Brazilian Amazon forest which has been recently highlighted byinternational press, (as it) plays an important role on the global climate equilibrium and on the global Greenhouse Gas (GHG) emissions, an important aspect of the Sustainable Development Goals (SDG). – checked

L55, suppress (it): With the Forest Code, (it) was created the CAR => give the meaning of CAR and function (not only to monitor deforestation, but map existing land use) – checked: we added to the description ‘forested areas’ as the CAR does not map land use, only the declared forested areas

L86: 69to 11.1km2 => 6.9 and misses thousand!!! – checked

We are sorry for the inconvenience with the minor English mistakes, the professional reviewer was contracted before these last changes and some details may have passed by our attention.